# Exploring the Role of Community Self-Organisation in the Creation and Creative Dissolution of a Community Food Initiative

**Mustafa Hasanov** * **, Christian Zuidema and Lummina G. Horlings**

Faculty of Spatial Sciences, University of Groningen, Landleven 1, 9747 AD Groningen, The Netherlands;
c.zuidema@rug.nl (C.Z.); l.g.horlings@rug.nl (L.G.H.)
* Correspondence: m.h.hasanov@rug.nl

**Abstract:** Community food initiatives are gaining momentum. Across various geographical contexts, community food initiatives are self-organising, providing communities with inspiration, knowledge and the opportunity to work towards responsible and socially acceptable transformations in food systems. In this article, we explore how self-organisation manifests itself in the daily activities and developments of community food initiatives. Through the conceptual lens of community self-organisation, we aim to provide a more detailed understanding of how community food initiatives contribute to broader and transformational shifts in food systems. Drawing on a multi-method approach, including community-based participatory research, interviews and observations, this article follows the creation and creative dissolution of the Free Café—a surplus food sharing initiative in Groningen, the Netherlands, which in the eye of the public remains unified, but from the volunteers' perspectives split up into three different initiatives. The results suggest that community self-organisation accommodates differing motivations and experiences embedded in the everyday collective performances of community rationalities and aspirations. This article also points to the changing individual and collective perspectives, vulnerabilities and everyday politics within community food initiatives. This paper contributes to emerging debates on community self-organising within food systems and the potential of community initiatives to promote broader social realignments.

**Keywords:** community self-organisation; collective action; community food initiatives; food activism; collective practices; citizen collectives; food sharing; food surplus; food waste

---

## 1. Introduction

This article explores the role of community self-organisation when considering community food sharing initiatives. While queries for understanding activism within food systems [1,2] and food sharing practices [3–5] are well-established, the rhetoric of community self-organisation within community food initiatives is relatively scarce. Over the last few decades, the literature addressing processes of civic engagement within the broader strand of community action and grassroots initiatives has been growing. Terms such as participatory society [6,7], big society [8], energetic society [9], active citizenship [10–12] and new acts of (urban) citizenship [13], among many others, have become popular. Such terms often convey innovative views on how to understand and analyse processes of community mobilisation, inclusivity and self-organisation. While community self-organisation is a trend amongst researchers and practitioners, it is less understood through the lens of the community members [14]. As such, community self-organisation appears to be a novel multi-individual and cooperative approach combining different motivations, norms and (dis)engagements. In this paper, we construe community self-organisation within the context of community food initiatives.

Community food initiatives are creating social spaces [15], which manifest notions of social engagement and social bonding [16,17], working together [18], food citizenship and civic engagement practices [19]. They promote alternative forms of food production and distribution, including food sharing and, in particular, the sharing of surplus or discarded food [20]. The main aim of this article is to explore how community food initiatives through processes of collective action can generate and fuel processes of community self-organisation, which contribute to a broader, transformational shift in food systems. While previous work has highlighted the need for a more radical reading in exploring how food initiatives connect with new forms of governance [21,22], this article focuses on how community self-organisation shapes the often-precarious existence of a community food, or in fact any kind of, initiative. A nuanced understanding of community self-organisation will be constructive for two main reasons. First, discussing community self-organisation helps us understand how impulsive collective action constitutes an expression of creative, dynamic and engaged citizenship, informed by a rich array of related and desired societal changes. Second, discussing community self-organisation highlights the variegated nature of community initiatives and their underlying ambitions, which inform us about how formal decision-making processes might engage with the variety of encountered community food initiatives.

We argue that such research is much needed, as it addresses a research gap in the context of the wider debate on self-organisation and social change. A key question in this debate is how citizen initiatives can be agents of change. To discover their potential, we need to see the detailed realities of these initiatives, not by observing them from a distance as a cohesive entity but by diving into their day-to-day practices, actions and rhetoric to find out how they self-organise. Our hypothesis is that (food) initiatives might not point at a unidirectional path of self-organisation but are seeking more diverse avenues from which to draw inspiration, as they consist of a plurality of cultures, relationships and identities. These pluralities have consequences for the ascriptions of collective intentionality, which are regularly attested as the outcomes of collective actions, in terms of the institutional arrangements, visions and activities developed by them.

This article explores such processes of collective action through community-based participatory research on and analyses of the creation, development and creative dissolution of the Free Café—a surplus food café and food sharing initiative established in the city of Groningen, the Netherlands, in 2014. Groningen is a distinct case for several reasons. To begin with, the city government is, within the Dutch context, traditionally considered left-wing and fairly progressive, creating a relatively favourable social and political context for community-linked initiatives to emerge and flourish. This is well documented in the field of rural community development [23,24], energy and sustainability transitions [25–27]. Furthermore, the city is surrounded by a large agricultural hinterland allowing for plenty of food initiatives connecting local communities with local or regional producers [28,29]. In this respect, this study aims to highlight the potential of the city and region within the bigger picture of urban food initiatives in the Netherlands [30–32]. In being a somewhat distinct case, there are certain limitations to generalisability of the research findings. As the main ambition is to disentangle how self-organisation is manifested in the actual day-to-day realities of initiatives, this research prioritised uniqueness and in-depth insights rather than generalisability.

## 2. Building Bridges between Community Self-Organisation and Community Food Initiatives

### 2.1. Community Self-Organisation

Existing research suggests that community food initiatives often have the characteristics of diverse economies [33] and set in motion various ideas, such as food alternatives and food diversities [34], connecting producers and consumers [35], autonomous food spaces, sustainable placemaking [36] and the creation of space for sustainable food systems [37]. There have been attempts to categorise different trends within food movements [38]. Authors have drawn attention to the potential of community food initiatives regarding different aspects of working together in building cooperative food systems [18],

politics of foodstuff [21,39,40], tackling food waste reduction [41,42], promoting sustainable food consumption models [43] and tracing food connections between consumers and producers [44]. In this context, community food initiatives provide an opportunity for people to pursue a common goal through means of collective action, solidarity and community self-organisation. It remains unclear, however, how community initiatives and local efforts concerning food might generate pathways and capacities to fuel a transformation of food systems through activism, cooperation and collective action. Within the discussions on emerging forms of food citizenship [19] and how individuals think and act differently while engaging in food initiatives [20], community self-organisation serves as a promising vehicle to outline how these initiatives lead to new social arrangements, public awareness and pathways for change.

Within the literary tradition of urban research and practice, self-organisation emerges as an umbrella term for a whole spectrum of activities. These activities include participation, inclusivity, "voluntarism, neighbourhoodism and informal or 'natural' support networks" [45] (p. 464). Public and academic discourses on self-organisation often, but not exclusively, refer to terms such as big society [8], participatory society [6], entrepreneurial citizens [46], do-it-yourself activism and experimentation [47], grassroots mobilisation and innovations [48] and mediating between lived experiences and formal entitlements in redefining acts of citizenship [13]. In turn, self-organisation reflects dynamic forms of collective action and community engagement across socio-cultural spheres that might have an impact on shaping—in our case—civic food networks, food citizenship and sustainable communities at large.

Embracing self-organisation urges us to reconsider the ability of civil society to set up and maintain community initiatives [49] in the context of existing institutional and governance structures [25]. Some authors speculate that there is a relationship between the growth of various new civic initiatives and the lessening of state and market dependencies [50,51]. As such, self-organisation can be easily seen as a convenient answer to governments and private organisations refusing to take their responsibilities [24]. Alternatively, self-organisation might also carry transformative tendencies helping citizen initiatives to trigger change in governance and policies. Scholars have also actively pursued the question of whether newly emerging citizen initiatives are successful or not [23,52]. In this context, self-organisation has become an apt descriptive tool to conceptualise how citizen initiatives develop and organise. However, while rhetorically attractive, self-organisation can easily hide the black box of actual decision-making processes and the ideas, ambitions and actions that drive them [53]. We know less about how self-organisation anchors and articulates different subjectivities within communities. By exploring the experiences of different individuals and the perceptions that exist within communities, we may develop our understanding of how citizen initiatives envision and pursue novel collective practices that aim to meet social needs.

Community self-organisation provides valuable insights into how collective actions are socially anticipated and loaded with rational, moral and intentional dimensions nested in a certain spatial context [54]. It shows how localised community-based projects can generate possibilities for collective action at multiple scales of interaction, allowing communities and societies to transform [55]. Self-organised initiatives mirror societal dynamics that highlight the discrepancy between small, localised project-led grassroots initiatives and the changing role of the state [56]. In this respect, most accounts on community self-organisation rely on relatively unproblematic notions of activities conducted in the absence of state, governmental or administrative control. That is, community self-organisation is conceptualised without including the interactions between the internal dynamics of the initiative and its wider institutional and social environment [25].

In response, this article opts for a more critical look at the actual processes behind how community initiatives organise and develop in taking actions. Notably, if we are to understand how they might fuel processes of sustainability transformations, we also have to understand whether the participants actually recognise this role and, if so, how they view their role. Currently, much of the existing literature assumes that it is clear which ambitions initiatives have and, thus, which actions these initiatives pursue. This suggests that the actual dynamic interplay of various participants and their values, views

and ambitions are overlooked, while the question of how collective intentions and possible roles are self-organising in this interplay also remains unanswered. Investigating how a diverse group of people engages in an initiative, works around their similarities and differences and gets something done could disentangle this interplay.

*2.2. Everyday Politics and Collective Action*

Instead of engaging with lengthy reviews of what is political, apolitical or post-political, in this article, we pursue a vision of everyday politics [57] within community initiatives. Everyday politics often revive sentiments of community activism, community organisation, visions of social change and the pursuit of social change and reiterate the principles of community initiatives at large [58]. Fischer et al. [59] suggest that the everyday matters of community-based or grassroots initiatives conceal different degrees of politicisation, the tensions between underpinning discourses and the organisational form and, hence, relations between what happens within the initiative and their wider social and institutional contexts. To understand how community self-organisation can lead to a broader influencing of social discussions and institutional practices, there is a need to acknowledge the polysemic and political nature of community initiatives. Hence, if we are to understand how those initiatives articulate multiple aims and pursue actions, we should unravel how a community or specifically a community initiative is understood, approached and endorsed [60,61]. The dogmata of everyday politics could provide a more detailed account of the pressing day-to-day issues relevant to community initiatives. It could also provide insights into the credo that underpins community self-organisation within the initiatives. In addition, everyday politics can account for the way in which initiatives are represented in formal, semi-informal and informal forums. This will help us understand how community food initiatives could influence and relate to wider sustainability transformations.

The political impulse of community food initiatives is of particular importance, as it can illuminate how community-bound social processes materialise and deliver sustainable, resilient and socially just developments. The political character of community food initiatives and food activism at large is worthy of attention, as it illuminates the creation of alliances and coalitions within food systems [1,39]. This suggests that community food initiatives are not the result of a simple summation but rather build upon diverse motives, identities, aspirations and actions for social change. Some authors suggest that community food initiatives express notions of democratic citizenship and radical political practices [62]. Others link food initiatives to issues of injustice, inclusion, exclusion and politicisation [21]. Within a broader perspective, such initiatives can be situated between "small, every day, embodied acts, often of making or creating that can be either implicitly or explicitly political in nature" [63] (p. 221), as well as acting on behalf of a cause. These studies acknowledge the everyday situations within food initiatives, which are essential in understanding how people with different motives, intentions and knowledge come together and—collectively—make something happen. While this concept could underline the notion of the collective within the initiative, it can also be a vehicle to assess the diversity and the potential impact these initiatives might have. After all, they are likely to pursue multiple paths that might trigger societal changes, rather than follow a single path.

## 3. Methods and Data: Exploring Community Self-Organisation

As stated in the previous section, understanding community self-organisation would require us to look closer into the developments, dynamics, opinions and discussions within community initiatives. Doing so, this empirical research is based on a multi-method approach to study these aspects and how self-organisation takes place within a particular community food initiative—the Free Café. The Free Café was broadly defined as a food sharing initiative and, more specifically, a food waste café, organising bi-weekly community meals in Groningen, the Netherlands. It started in late 2015 as an art project with a loosely organised structure, with low or non-existent levels of institutionalisation. The Free Café explicitly and implicitly carried environmental, social and political undertones in its operation. It also provided insight into how community self-organisation takes place

within food communities, or, in fact, any initiative. The data presented in this article are drawn from a multi-method research design, including community-based participatory research (CBPR). Falling under the umbrella of action research methods, it focuses on positive social changes in a particular community through the involvement of community members and researchers [64]. The rationale behind adopting CBPR is that it emphasises the "participation, influence and control by non-academic researchers in the process of creating knowledge and change" [65] (p 183). It provides multiple ways of knowing what is happening inside communities by recognising "community as a unit of identity", "building on strength and resources within the community", "integrating knowledge and action for mutual research", "promoting co-learning and empowering process" and "disseminating findings and knowledge to all partners" [66] (pp. 178–181). In the long term, CBPR provided valuable insights into the day-to-day operations and issues of importance to the Free Café. Alongside CBPR, this research also incorporated qualitative research methods such as semi-structured interviews, field notes, observations, interpersonal and group communications and other information sources. The rationale for choosing a multi-method tactic is due to its ability is to provide analytically salient data triangulation and rigour [67].

Qualitative methods are suitable for accessing more in-depth information, such as how volunteers give meaning to their activities and how the day-to-day activities of initiatives happen in order to peek into the internal world of the initiative. In the period between April 2015 and December 2016, the corresponding author was actively involved in the Free Café, mainly responsible for minute taking during meetings and as an occasional kitchen volunteer. To ensure data were clear and reproducible, he made use of a digital research diary to keep track of research activities, thoughts and feelings after, and sometimes during, each visit to the café. These notes and observations provided insights into the following: how the initiative incorporated knowledge gained through experience; what the logic behind specific interventions was; how its ambitions, actions and possible influences on wider social publics changed; and how the health, quality and public perception of the initiative fluctuated over time. To avoid or reduce bias that might be associated with the author's closeness to the initiative, this research also included other sources. Hence, the data collection included 30 semi-structured interviews with core team members, regular volunteers and visitors of the café. Each interview included themes such as mobilisation and motivation for setting up or being part of the initiative, ways of organising, reproduction and development of actions and practices, the role of kinship and friendship within the initiative, working with others and key challenges, hurdles and everyday practices at large. To ensure that the full picture would be larger than the insights taken from active participation and interviews, the data collection also focused on interpersonal communication during volunteering activities. Additional information sources, such as including news items, videos and social media discussions, provided a complementary perspective to primary data sources. In sum, the multi-method approach provided variegated information on the essence of the dynamics of the initiative and its connections with other initiatives and local authorities. For the most part, it also allowed us to capture the volunteers' attitudes regarding the way in which they view their initiative and also provided information on the role of the visitors, other initiatives, the private sector and the government, from the bottom up.

Data retrieved in the frame of this research was digitally recorded, transcribed and rendered anonymous following confidentiality and data protection guidelines. The multi-method approach produced different data types that complement and supplement one another. This required rigorous analysis, which was performed using Atlas.ti (version 7.5, ATLAS.ti Scientific Software Development GmbH, Berlin, Germany), a computer-assisted qualitative data analysis software package. This research combined a theory-driven approach to coding with a more inductively informed approach. Some initial coding themes referred to particular notions of collective intentionality, self-organisation, motivations, aims and goals, location, internal and external community politics, ways of organising, tension points, representation and perception of the initiative. In the process of collecting data, additional codes were adopted, such as community identity, *free* food vs. free *food*, learning, reflecting, food access, exclusions and food networks.

A note should be made here concerning issues of the authors' positionality and the role of the researcher in doing research [68]. The data reported hereafter risk being treated as 'biased', as they represent insiders' perspectives of the initiative, because the corresponding author was also part of this view. However, the engagement of the corresponding author in the Free Café built trust and established rapport with community members. It was clear from the beginning that the corresponding author was not only a volunteer, but also a researcher and thus collecting data. Volunteers were aware that they would reveal and share potentially personal and sensitive information. At the same time, the corresponding author took a background position, which meant that the corresponding author did not have an active stake in what was happening and being decided on in the everyday matters of the initiative. This allowed to the corresponding author to appreciate and acknowledge the different situations and views present in the initiative. It also allowed the corresponding author to see the how these changed over time. The combination of multiple sources of evidence also allowed us to see the Free café through multiple perspectives and enhanced the credibility of the data.

In working with local communities, it is essential to be aware of the unanticipated risks and events regarding the direct engagement of the author. This was particularly evident in this research. Effectively, when the data collection commenced, there was a single initiative, but when the corresponding author was retrieving data from the field, three smaller initiatives had formed. While unforeseen, such development provided an opportunity to observe an additional viewpoint and explore unanticipated (self-organised) reactions to different strategies among volunteers regardless which initiative they supported. It is equally important to be aware of the potential dangers of using multiple methods and being actively engaged in the selected case. The corresponding author kept low profile, but this does not mean that his involvement was not noticed. Some of the volunteers and visitors were uncomfortable with the fact that the corresponding author had a dual role, and others simply avoided contact with the corresponding author, as they were not sure in which capacity he was acting at a given time. While this did not influence the quality of the data, it is indicative of the intricacies related to participatory research. Despite his participation, the corresponding author had no influence over the development of the initiative. There were two exceptions. The first one was when the researcher had to interfere in and pacify an argument between some volunteers during a general meeting. The second one, although indirect, was when his name was mentioned in a bid book for a new location as a 'researcher', and this added external validation to the permit application. Although it is difficult to assess whether including the corresponding author's name in the bid book influenced the outcome of the application, it emphasised the role of socially engaged scholarship in relation to community self-organisation. These examples illuminated various said and unsaid signs and constructions situated behind the development and disestablishment of the interpersonal relationships and power dynamics hidden in the dynamics of community initiatives.

## 4. The Free Café: Sustain, Flip or Let Go

We begin this section with an overview of the Free Café and how it developed and creatively dissolved. This overview is primarily based on documentation found on websites, direct observation through the participation of the corresponding author and interview data regarding how those involved believe the initiative started and developed. Then, we will address how self-organisation and collective intentionality are perceived by community members and how these concepts have manifested themselves in the Free Café. The discussion on these perceptions is based on the analysis of transcripts of the interviews and relates to the categories of codes relating to both concepts. These discussions are complemented by the observations of the corresponding author during his involvement in the discussions and the everyday practices of the initiative. This involvement was also the prime source of data to better grasp how the initiative and the debates within it evolved. Hence, the observation of discussions and the gradual development and creative dissolution of the initiative into three distinct initiatives also helped to establish a more factual narrative of how both concepts manifested in practice. This section ends with a more reflexive overview of the dynamics of interaction occurring during

the development and creative dissolution of the initiative, which helps to explain how collective intentionality and self-organisation are drivers for change within the initiative.

### 4.1. The Free Café: A Brief Overview

The Free Café was a food waste café, established in late 2014. It was committed to cutting down food waste by preventing it by using edible food that would otherwise be discarded. The initiative started as an art project that aimed to create an inclusive environment where everyone can find his or her place. Therefore, as was also expressed during almost all the interviews, food sharing was an ideology that held together different ends of the volunteers engaged in the initiative. The volunteers collected surplus food from various sources, such as local markets, supermarkets, farmers and sometimes the local food bank. Meals prepared from waste food were cooked and served twice a week. According to the initiators, the Free Café operated on a voluntary basis, relying on generosity, collaboration and, more importantly, avoiding the use of cash incentives or monetary exchange. There was no register of volunteers, although there was a core group of volunteers and regular helpers. Visitors were also welcome to help. As one of the initiators put it simply during a meeting, "There [was] a low threshold to become active, and this [could] be done at one's own pace or manner". The crowd attending the meals was consistent, consisting of students, creatives, people with part-time jobs, senior citizens and sometimes people facing difficulties in life. During various interviews and weekly organisational meetings, it was mentioned that some associations assisting the homeless and people in need also recommended that their regular visitors attend the café.

Positive public exposure and attention in media, as well as word of mouth and social media posts, promoted the initiative between November 2014 and June 2015. This caused a surge of visitors and volunteers, which according to most volunteers led to in-group tensions regarding the future of the initiative. From the beginning, the café was hosted by another local initiative, which had to relocate by the summer of 2015. It was made clear that the Free Café would not be allowed to relocate with its host. In September 2015, the Free Café was evicted, and during a general meeting regarding the fate of the initiative, the volunteers decided that the café could not continue to exist as a single entity. This decision, along with other emerging personal clashes and ideological differences between the growing number of volunteers, led to the volatile split of the initiative. In early 2016, the corresponding author witnessed through observations and occasional site visits that pioneering volunteers started to work on 'De Wandeling' ('*The Walk*'). According to their Facebook page and website, this is a social initiative with broader ecological and environmental goals, aiming to create an environment where everyone feels welcome and free to join. This subgroup wanted to create a piece of wonderland where people can enjoy peace and nature by showing that it is possible to build with materials that are in abundance. Additionally, it intended to promote workshops where everyone can learn the techniques, develop the skills and gain the experience to optimise their participation in the initiative.

At the same time, the volunteers who shared the goal of reducing food waste continued to organise the weekly meals in different locations. In the autumn/winter period of 2015, after what could best be characterised as a small identity crisis, the Free Café was renamed the Free Cafés. In May 2016, the Free Café@Backbone opened its doors. For a short period, it was called "Restaurant Restant", which carried the connotation of being a social eatery where visitors could enjoy food prepared from surplus food. The relatively unpopular location, expressed in interviews with volunteers, participatory observations and interpersonal communication with visitors, raised some concerns, and soon another spin-off took place at a more easily accessible location. The third sub-initiative, the Free Café@Edanz emerged in late 2016. While the aim of the Free Café@Edanz was similar to Backbone—sharing a meal from designated waste or surplus food—the ways of organising the initiative were different. The Free Café@Edanz relied on a more professional approach, and there was an allocated manager responsible for food collection and the preparation of the meals.

### 4.2. Perspectives on Community Self-Organisation

Arguably, the café adopted multiple and sometimes incompatible perspectives, which can be seen not only in how the initiative changed over time, but also in how volunteers were relating to what they thought the Free Café was about. Those perspectives were not fixed or static; they morphed over time, as the composition of the café depended on the volunteers participating in the activities and their experiences. Based on the aims and motivations expressed by volunteers, three perspectives on community self-organisation could be distinguished. Based on interviews, observations, attending various meetings and public representation of the initiative, the following perspectives were distinguished: (i) a critique against neoliberal practices concerning ecological activism; (ii) social interaction, partnership and mutual trust; and (iii) the transformation of society and the current economic system based on resource allocation and environmentalism.

The first perspective illustrates a frustration with current political, social and environmental processes and an opposition to neoliberal practices. The goal is to reform the entire way of thinking about sustainability and provide accountable and fair alternatives to capitalism. This perspective suggests that the Free Café stood for redefining wellbeing, living together, providing alternative notions of sustainability and pursuing free behaviour and rationality. Johnny, a regular volunteer suggested that the Free Café had something to do with showing that it is possible to work together and in solidarity.

> *"Because a free cafe, a cafe where everyone meets, where everyone is equal, where no hierarchy is existing is something that really supports the social dimension [of the café] and equality."*

His outlook is partly observable in the second perspective, according to which it is imperative to build alliances and coalitions with semi-public and private bodies in pursuing collective goals that will transform food systems. In taking collective action, the development of mutual trust, partnership and togetherness is essential, as well as to strive for social inclusion. According to Carola, who helped to coordinate the cooking process and was an experienced volunteer, there should be some balance between initiatives and other parties.

> *"I am not against cooperation. If there can be cooperation, it is good. If there has to be confrontation, well, confront. But if there can be cooperation between government, the scientists, corporations even; if they want to cooperate, it's all right."*

These words resonate moral and ideological convictions, but also prioritise getting things done. Another aspect of the café, as reported by Manuel, an occasional visitor, was related to raising awareness through elements of pleasure

> *"Even though people go there for fun, there's this, we're talking about collective consciousness, there is this movement going on in people's mind[s], which is saying 'we want it [things done] in another way'."*

The third perspective is observably the least radical of all three. It confirms commonly accepted norms and rules and disagrees with the idea that community initiatives, such as the Free Café, point to the limitations of local governments. While agreeing that humanity should live in harmony with nature, it disagrees that the café was channelling environmental concerns. It instead prioritises the importance of acting together, belonging to a cause and avoiding confrontation. Agnetha, who volunteered in the café since its creation, suggested that while it is easy to slip into anti-neoliberal critique and anarchist thinking, at the same time we also need law and order

> *"Some sort of government is always needed . . . what I like to think is this the responsibility of the government acting in the public interest. If it [the government] is doing it efficiently that's a bit different."*

Hence, this perspective asserts that inclusive collaboration and cooperation between the Free Café and other public actors was vital.

The data collection, and especially the participatory observations and interviews, commenced when the Free Café was one big initiative and was completed when it had become three different initiatives. In this conversion, the corresponding author witnessed not only how some volunteers dropped out and new ones arrived, but also how the identity of the Free Café changed incrementally over time. Retrospectively, it was seen as a refuge from current neoliberal realities, addressing various aspects of freedom, ethical and moral action. The Free Café harnessed the power of imagination in light of togetherness, feeling good and embracing environmental ethics. Lastly, the initiative appeared to be less radical, with an outlook incorporating entrepreneurial logic, in which the Free Café was a temporary place of consumption and resourceful activism. Those three different perceptions were also the starting point of the split in each sub-initiative. Nevertheless, it was difficult to determine whether each perspective was exclusively kept only within one sub-initiative. These perspectives were constructed via social interaction and changed over time. That was visible especially in the case of the Free Café@Edanz and Free Café@Backbone, where, for a long time, there was uncertainty as to whether to partner with mainstream organisations or remain as an independent initiative. Arguably, within each sub-initiative, one perspective was more dominant than the others, while not excluding the other perspectives. Addressing and understanding the changing relations between these perspectives required a more in-depth and qualitative analysis of the different situations that emerged in the everyday experiences within the initiative, which we summarise below.

### 4.3. Everyday Collective Experiences

The Free Café's vision was a nearly utopian and ideologically driven community-led project-based endeavour from the beginning. Aspects of inclusion and equality were the intangible values on which the initiative focused, while the food was the tangible reflection of those values. In a group discussion, some volunteers mentioned that the right word to describe how the Free Café worked was 'nice-ocracy', as some of the volunteers called it—the rule of niceness. This can be understood as an augmented version of sociocracy where decisions are made in delegated horizontal structures through the use of consent rather than the opinion of the majority. Surplus food was the underlying element which defined the initiative, but the Free Café was not just about free *food*: it was about *just* and *free* food. It stood for social and ecological justice, self-determination and self-reliance. To some volunteers, the issues of food were central to the initiative: the fact that they could walk into the café, share a free meal and have a conversation with each other. Pernilla, one of the pioneering initiators, mentioned that food was a very familiar topic for everyone who wanted to get involved in the café.

> *"One of the focus points is food. Because it's a really interactive part, and it's really easily accessible for everyone."*

To others, such as Margaret, another long-term volunteer, the origin of the café was associated with feelings of personal and social freedom or liberty and existence without undue or unjust constraints.

> *"[T]he point is that we want to facilitate sustainable ways to meet human needs that operate on the basis of generosity, ingenuity, creativity and cooperation . . . and also encourage free exchange of ideas, knowledge, and skills."*

The dichotomy between the materiality and immateriality of the Free Café was a reason for heated discussions on the true essence of the café. A very practical example of this dichotomy was an ongoing discussion about whether or not to use fossil fuel vehicles for food collection. Sometimes, the café would receive a notification from a farmer who had surplus harvest. It was only logical that a volunteer would organise a car to pick up the free produce. Some volunteers disagreed with this practice, as it was polluting the environment and against the principles upon which the café was created. Those disagreements usually occurred behind closed doors, and sometimes online, but were quickly resolved.

Some happened during opening hours and caused some of the more vulnerable visitors to leave. The moral of these, sometimes heated, debates was that the Free Café was not ideal, but it was a place where we could learn lessons.

Another source of discord was the collection process of waste or surplus food. Collecting the food was not an issue most of the time, but it had some intricacies. For example, when food collection became a regular weekly activity, some market retailers began to designate waste crates for the café. However, there were reports that people who rely on the market surplus were denied the discarded produce of the day and referred to the café. During interviews, some volunteers shared that they felt uneasy that other volunteers would collect an individual share of surplus food, along with the items designated for the café. In the long term, ideological and moral differences were less substantial than the fundamental incompatibility of volunteers' personalities, approaches and lifestyles. As became evident from the data collection, there was not a single but rather several micro-communities within the Free Café. Each of them had different visions about the future of the café. Eventually, those micro-communities formed the core of the sub-initiatives outlined earlier. In this divided situation, some volunteers preferred helping De Wandeling start up and become popular, while others continued to collect surplus food and offer it for free. The two helper groups interacted with each other only when necessary. In post-fieldwork communication with some of the volunteers, it was discovered that some helpers withdrew from the Free Café@Edanz, for example, as they felt uncomfortable with how things were "happening" at particular locations. Similarly, some visitors felt that the new locations did not speak to them or represent the "coolness" of the Free Café.

*4.4. Dynamics of Interaction*

During various conversations and in traditional and social media reports, the café was romanticised by volunteers and also by the public, often being depicted as an artistic refuge for free food, alternative lifestyles, do-it-yourself crafts, happiness and care. The nature of the initiative, volunteer influx and turnover and varying levels of volunteer engagement revealed an interesting picture of the dynamics of interaction within the café and between the café and the outside world. Drawing on observations and conversations, the corresponding author found out that most of the long-standing volunteers had knowledge and experience or had built a network that was relevant to the Free Café. Some of those networks were pre-existing, some were created within the initiative itself, some were based on previous volunteering experience, some were based on education, some were based on paid work and some were based on pure intent to do something meaningful. For example, the core group of volunteers, which later set up De Wandeling, knew each other already and worked on various art projects prior to the café. Hence, they knew how to put things together and already had a network that was useful in setting up the initiative. Some of the people who formed Free Café@Edanz and Free Café@Backbone knew each other from other volunteering in other community projects. The kitchen crew were generally educated and had experiences in cooking and catering. Some volunteers who were experienced in public exposure and outreach were frequently approached to present at local and regional events, give interviews for thematic magazines and participate other public activities. The initial hype put pressure on the limits of what the initiative could deliver. Not only was it romanticised by the media, but also volunteers and visitors found the intimacy and character of the café to be one of its essential features. Multiple news items and social media posts focused on the almost-utopian character of the initiative and the unique atmosphere it created.

On the other end of the spectrum, local authorities framed the Free Café as a social initiative within the larger frame of 'citizen participation' and 'caring society'. Gradually, some links with a tolerating government were established. As such, local authorities did not provide, nor deny support to the initiative, but would just let the initiative be, thus taking on a passive role. Simultaneously, the café developed close ties with other voluntary initiatives or non-profit organisations, such as the local food bank, regional environmental protection organisations and similar food-related initiatives. Those networks were useful when the volunteers were determining how to continue working on De

Wandeling. The initiative was also creative in the ways in which it presented itself to authorities. The official planning application for De Wandeling stated that within the core team of volunteers there was also a researcher (the corresponding author), emphasising that the initiative was not just another citizen project, but that it also fostered co-creation and learning with regard to sustainability. The Free Café changed its "behaviour" from proposing or advocating impractically ideal social and political schemes to more pragmatic methods, which were somehow still ideologically inspired.

## 5. Discussion

With respect to the aim of this research, the findings outlined above highlight three important dimensions regarding community self-organisation. The first dimension refers to the conscious creation of ties and how collective actions within initiatives materialise, which might potentially be applicable to other food initiatives. The second dimension covers the interactions between and within community-based collectives. It reveals that community self-organisation is a state of mind that might lead to unexpected outcomes. The third dimension refers to the horizontal engagements amongst volunteers within community initiatives, showcasing how people in a specific setting deal with differences to achieve something.

### 5.1. Community Self-Organisation and Community Food Initiatives

The findings of this research have repercussions both for understanding the role of community initiatives in transforming food systems and for understanding community initiatives themselves. In practice, ideologies and ambitions regarding how to deal with food in a sustainable and fair way were directly linked to ideologies on fairness and sustainability in general. To start with the relevance of free café in the context of the wider debate on food regimes and movements, our observations resonate with different components of the dominant food regime and food movement frameworks reported in earlier studies [38]. Drawing on the analysis of the data and active engagement in the field, it is not difficult to see how community food initiatives, such as the Free Café, contribute to the corporatisation and commodification of surplus food [2]. We identified multiple similar initiatives focusing on the prevention of food waste in a regional, national and international context. While those initiatives emphasise the authenticity of surplus food, they also stress the importance of food sharing as a binding element for wider goals and visions [69]. However, some aspects of the initiative were clearly more "food" oriented. The meals prepared from leftover produce were aiming not only to raise awareness and educate people on the matter of food waste. They also illuminated the importance of short food supply networks [19], personal and social reconnecting to food [44] and sharing as a strategy to tackle food waste [42]. Food sharing was essential for the café in order to make sense of and question contemporary food politics [39]. At the same time some of the benefits of shared dining included nutritional education and promotion of healthy diets. While these findings relate to earlier research [5], one should keep in mind that perceptions of nutrition and health are subjective and depend on lifestyle choices and people's diet [70]. Some of the findings also resonate with earlier research on the economic, environmental and social benefits associated with sharing practices with regard to food waste reduction [41]. However, within the initiative, food sharing was framed in terms of "freedom" in a society that is becoming more unstable, more anxious and more uncertain, where an individual can choose and shift between different social positions [71].

Driven towards utopian states of sustainability, ecological societies and solidarity, a case such as the Free Café also shows that how ideologies and ambitions regarding transforming food systems were directly linked to ideologies on fairness and sustainability in general. Hence, the initiative also indicates how community self-organisation has the potential to promote social innovation by building new social relations and to work towards social change [72]. Similar to other studies, we also found out that creativity and innovation are an integral part of food system transformations [2]. It is likely that local cultures and practices are essential in understanding self-organisation and the way in which these might contribute to such transformation.

The uniqueness of this case makes it difficult to assess how generalisable the results are. However, the Free Café, as well as its successors, provides us with insight into what community-driven interventions can achieve. According to our data, we would argue that the rumblings of discontent and overcoming self-interest within community initiatives allowed and maybe even triggered new initiatives and innovations to emerge strategically. While contested within the initiative, the move towards professionalisation and the established agreements with actors such as supermarkets and markets suggest that the Free Café began to change the local food system. Clearly, a small project-based community food initiative does not necessarily indicate a trend, nor can it account for a complete shift of the food system, supporting a more sustainable, resilient and socially just development. Nevertheless, initiatives as the Free Café can play a role in indicating the direction of the broader systematic changes needed for transformation. As the evolution and creative dissolution of the Free Café suggests, they can at least be starting points for such systematic changes. In doing so, the Free Café provides an example of transforming food systems through the work of community initiatives.

## 5.2. Vulnerabilities and Creative Dissolution

The second feature in the analysis relates to the importance of vulnerability within community initiatives. There is a tendency in the existing literature to make a distinction between success or failure of grassroots initiatives without gradation [23,52]. Here, a more nuanced categorisation might be useful, namely one that takes into consideration the dynamics and change of initiatives over time, which can often be confused with failure. In this case, the initiative creatively dissolved into three distinctive community projects. None of the spin-offs harboured negative feelings towards their predecessor, and the café was rarely discussed within framework of failure. Indeed, some of the volunteers withdrew due to a state of emotional and physical exhaustion caused by a prolonged period of stress and frustration, and some respondents expressed their disappointment that the Free Café was not what it used to be. However, this research reveals that failure and vulnerability are open to interpretation and, more importantly, depend on the capacity of volunteers to collectively cope with or escape pressure. Regarding the three different perspectives outlined in the previous section, we saw that the perceptions of success and failure differed greatly between people. In other words, the notions of failure and success were beyond the initiative itself. Thus, it became important to operationalise and rationalise different aspects of influence, values, social awareness, cohesion, institutions, new practices, changing meanings and the constantly shifting 'somethingness' of initiatives.

The volunteers might have recognized the development of two or three distinct initiatives, but from an outside perspective, the free cafés were still a single café. This shows the relevance of understanding how small localised projects forge forms of collective action that reach beyond the established ethea of the movement-building agenda. This is where self-organisation, with a renewed emphasis on civic governance, becomes relevant. Self-organisation involves the politics of community that focus on processes that are dependent on circumstantial conditions and interlocking arrangements of power. Such findings resonate with the governance logic of civic food networks outlined by Anderson et al. [18]. Furthermore, we also suggest that self-organisation stimulates a continuous process of transformation, which might have "unexpected outcomes". Sometimes it causes a series of fortunate or unfortunate events, which lead up to the creative dissolution of an initiative into sub-initiatives that comply with the particular requirements, conditions and alignments of the interests in question.

## 5.3. Everyday Politics

Our analysis also revealed the relevance of understanding the everyday politics of community initiatives. The Free Café required various qualities, capacities and talents needed to keep the momentum of the initiative going. The initiative provided space for the expression of collective action and alternative organisation within, against and beyond mainstream neoliberal practices. The volunteers and visitors of the Free Café engaged in critical discussions, created their own identities and solidarities through shared experiences rather than via political affiliations. The Free Café

demonstrated the existence of collectivist intent in community initiatives and how such intentionality can systematically challenge the rationalities of market and state. It also explored distinctions of individual and social action and how to create alliances, overcome differences and the importance of everyday knowledge together. As such, the café, through community self-organisation, blended personal interests and politics of community into one.

In the existing literature on urban food systems, new forms of organisation and politics have been used to describe a critique on the neoliberal regime and to cultivate new spaces of change [21,22]. The findings from the Free Café confirm this. The findings suggest that self-organisation within community initiatives contributes to imagining and befriending different options that might seem radical. However, this does not abandon the logic of conventional politics and regulatory frameworks. Self-organisation is somewhat cosmopolitan, provisional and situational. Looking at the everyday politics of the café and how internal differences lead to the rise of three initiatives, this study suggests that community self-organisation contributes to problem-solving and hence leads to the illumination of more pathways, more diversity in practices and more opportunities for social change. The three initiatives indeed point to different paths for change and thus show how differences do not always melt into a single collectivity, but that the collectivity can also disintegrate, pursuing various pathways for transformation. This points to the idea that self-organisation does not need to result in one pattern of change in a single initiative but might develop into different patterns based on differences within initiatives. More empirical research is needed, however, to confirm this statement.

This research builds on earlier reports addressing the nature of what is "community" and everyday politics of community initiatives [59–61]. The initiative was eager to achieve radical social and political changes, but it also relied on different everyday politics to continue its own existence [40]. The particular ethics and set of practices leading to the creative dissolution of the café in sub-initiatives illustrate well the importance of constructing and formalising such politics. However, the case of the café also indicates that the politics of community initiatives can be based more on kinship and friendship rather than on ideological versatility. The findings illustrate that the volunteers create and nurture some form of collective consciousness, which reflects a specific culture and heritage. This collective realisation is part of a shared cultural history, which is engraved into the initiative's blueprint. There are, however, other possible explanations. Our data suggest that volunteers collectively negotiate and deal with internal differences to get something done, which remains unseen to the public. The relations of the Free Café with local authorities and other tertiary sector organisations illustrate this point well. The public authorities saw little difference between the original Free Café, the Free Café@Backbone, the Free Café@Edanz and De Wandeling. The latter was eventually portrayed as a separate initiative but only after the initiators highlighted the differences. The public, including the local government, saw a unified initiative that simply collected food and organised community dinners. What started once as a utopian dream became a space for collective food consumption, while for visitors and observers nothing changed.

## 6. Conclusions

This article revealed how a community food initiative, through processes of collective action, can generate and stimulate processes of community self-organisation, potentially contributing to a broader change in food systems. The results also indicate that community self-organisation can contribute to emancipatory knowledge on the everyday politics within community initiative and how the dynamic interplay between intentions, perspectives and actions produce self-organised collective performances. The results suggest that community self-organisation should not only be discussed solely through a community lens, but also should be considered in terms of cooperation, collaboration and distribution of responsibilities within and between initiatives. While sustaining strategic alliances in day-to-day operations, these findings are in agreement with some of the earlier comments on food democracy and governance in civic food networks [19].

Empirically, this article examined processes of community self-organisation within the context of the Free Café, a surplus food sharing initiative situated in the city of Groningen, the Netherlands. The discussion helped us to understand how to approach and endorse community self-organisation in the specific context of food initiatives. Our case suggests that community food sharing is not only just about sharing food, but also about putting ideas into practice and imagining how food systems should be reformed to deliver sustainable, resilient and socially just results. The motivations and values of people within an initiative may follow different notions and trajectories of collective action as a result of societal narratives and discourses. Sometimes these can be complementing, and sometimes these can be contradicting, embracing creative dissolution. However, the dissolution of an initiative should not be interpreted as a failure. On the contrary, dissolution underlined the synergies and relationships between the different issues and opinions present in the initiative. In its essence, dissolution released various resources, capacities and social capital that became available for repurposing and that could be reassembled into new initiatives. Perhaps one can see the resemblance here with the notions of destructive interference and creative construction [73] depicting separate phases of a project. In this case, creative dissolution reaffirmed the point that social capital and self-organisation are closely interrelated.

This article also revealed that self-organisation is not just the exemplar of a participatory society. It discloses how different aims and motivations, community politics and representations of community initiatives are constantly redefined, negotiated and communicated, not only within, but also outside the initiative. Hence, neither community self-organisation nor community food initiatives are subject to fixed properties or qualities. Instead, they are in in flux. Whereas sometimes alterations could alarm hidden conflicts that led to creative dissolutions, in the long term such dynamics and fluctuations highlighted the capacity of community initiatives to change perpetually and offered complementary pathways for such transformation. The Free Café, placed in the heart of this research, aimed to create a refuge where people could freely share food, knowledge and experience. It was revealed that internally the initiative embarked upon different visioning and organising exercises that lead to what has been described as a creative dissolution.

Based on the aims and motivations expressed by volunteers of the Free Café, three perspectives on community self-organisation could be distinguished: (i) a critique against neoliberal practices concerning ecological activism; (ii) social interaction, partnership and mutual trust; and (iii) transformation of society and current economic system based on resource allocation and environmentalism. The described perspectives draw on ideas of socio-ecological justice, care, food access, exclusions and alternative food futures. We also found out that better understanding of community self-organisation calls for recognitions of vulnerabilities and community politics. Community self-organisation stimulates thinking and discussions about the many issues associated with community initiatives. Such issues signify the importance of interchanging issues of culture, relationship and identities, not only within the café, but also in what appears to be external to the initiative. Public discourses around the initiative reinforced and propagated a utopian outlook. People less familiar with the Free Café saw an initiative that prepared and served waste food twice a week in different locations. Such associations focus on community self-organisation as lived experiences, animated by values, cultures and intersubjectivities that need to be unpacked further.

Our findings also have repercussions for how media, market parties and governments might improve community initiatives. Some critical tensions within the initiatives occurred due to the different ways in which the initiative was popularised and could thus be professionalised. It is all too easy to assume that a community initiative has a singular identity and purpose. The reality can be quite different, with tensions upon what and how the initiative should be according to its participants. If society is to foster the transformative power of community initiatives, it should at least acknowledge the possible transformative seeds these initiatives hold. Pushing for only one distinct path, after all, can undermine many other transformative powers embedded in initiatives. For the local authorities, for example, the initiative was mainly seen as a social project, ignoring how it was also addressing

issues such as reducing food waste and sustainable food consumption at large. Similarly, for the media, it was mostly an exciting eco-hype, which triggered some participants to pursue a more commercial path and others to resist and focus on elements of sharedness and nice-ocracy. Finally, a key lesson is to accept that dissolution need not be judged negatively. A dissolution might sound like a failure to some, but every disappointment might well contain the seeds of future success.

Regarding to commitment to theory and explanation, several questions remain to be answered. The findings of this research suggest that community self-organisation, while being heterogeneous, reflects collective intentionality, which means that it involves in-process and goal-oriented attitudes within community initiatives. Collective intentionality highlights the skills, talents and motivations required to articulate the aspirations and needs of citizens in local initiatives. Whereas debates on collective intentionality are well-established [74] and relevant to grasping the formation of new civic consciousness within local initiatives [54], they rarely discuss the social infrastructures that local initiatives build. Collective intentionality is a signpost, which helps to reveal the different meanings, settings and dynamics of community initiatives. Discussing community self-organisation in the context of collective action can also help us to answer the question of how collective interest evolves into collective action exactly. Studies addressing self-organisation or community initiatives hardly acknowledge the dichotomy between individual gain and collective responsibility. We suggest, therefore, that future studies should be aimed at a better understanding of the performance of community initiatives, as we need to be careful of how we define the failure or success of an initiative. This research followed the success and creative dissolution of a community creation, the Free Café. To some it succeeded and to some it failed, only to be resurrected. After all, by doing exactly that, community initiatives are able to inspire other communities to develop, merge, dissolve and, notably, engage with sustainability-driven societal transformations.

**Author Contributions:** Conceptualization, M.H.; Formal analysis, M.H.; Investigation, M.H.; Methodology, M.H.; Project administration, M.H.; Resources, M.H.; Writing—original draft, M.H.; Writing—review and editing, M.H., C.Z. and L.G.H.; Supervision, C.Z. and L.G.H.

**Funding:** This research was part of the activities within the European Joint Programming Initiative on Climate (JPI Climate) project SELFCITY funded by the Netherlands Organisation for Scientific Research, grant number: 438-14-803. The views and opinions expressed in this article are those of the authors. They do not represent or reflect the views of the institutions that funded this research.

**Acknowledgments:** This research would not be possible without the kindness and generosity of the volunteers and visitors of the Free Café. Personal thanks to (in alphabetical order by first name) Elisabete Figueiredo, Gerald Aiken, Gvantsa Salukvadze, Luke Owen and Moya Kneafsey for their valuable comments on earlier versions of this manuscript. Naturally, the authors are responsible for the remaining errors, although, in our opinion, the reviewers and editors could have caught few more.

**Conflicts of Interest:** The authors declare no conflict of interest.

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
