# Peer review of "Exploring the Role of Community Self-Organisation in the Creation and Creative Dissolution of a Community Food Initiative"

_sustainability, doi:10.3390/su11113170_

Round 1

Reviewer 1 Report

The paper is interesting and it touches upon a timely issue, since grassroots food initiatives are gaining momentum across European cities. Nevertheless, some questions have to be taken into account in order to exploit its full potential.

The initial paragraphs (Introduction and par.2) are too long and very general, it would be more effective to shorten them and introduce the main hypothesis of the paper. At the same time, since the object of the paper is a very local and specific initiative, it should be put it in context: the authors should provide basic information about the diffusion and development of similar initiatives in Dutch cities, and about the context of grassroots initiatives (not just in the food domain) and active citizenship in Groningen. 

In par.3 on Methodology, the authors should critically reflect also about the limitations,  shortcomings, and problems connected to the combination of different methodologies adopted and to the situatedness and direct engagement of one of the authors in the practices analysed.

In par.4 a sound description and analysis of the actors involved and of their networks or 'communities' (pre-existing or created within the initiative itself) would help to clarify some issues related to the dynamics of interaction which, on the contrary, remain in the background. 

Finally, in the final discussion part, the authors should more clearly disentangle the food-related dimension of the initiative under scrutiny from other, community or interaction-related ones.

Author Response

Thank you for your kind words and constructive feedback. We are grateful for your review and insights on how to improve further this article. We do not doubt that having attempted to address your feedback point by point, the paper is much stronger

Following the recommendations of all reviewers, we highlighted (green font) some of the more critical corrections in the text. In the attached file you can find our repose to your comments.

Sincerely,

the authors

Reviewer 2 Report

The present paper investigates explores the role of community self-organisation when considering community foodsharing initiatives. This work is worth to be read and shows a good potential if several challenges are addressed:

Abstract has inappropriate structure. I suggest to answer the following aspects: - general context - novelty of the work - methodology used - main results

Introduction presents interesting information. It is really well structured and comprehensive. In the introductive section I would see a clear focus on the themes concerning food waste reduction and valorization as well as food sharing. Several projected dealing with it have been founded and a myriad of cutting edge papers have been published. Some reference to start with, but not limited to, are:

Valorization, reduction and sustainability:

https://doi.org/10.1016/j.jclepro.2018.01.208

https://doi.org/10.1016/B978-0-12-811935-8.00011-1

https://doi.org/10.1002/bbb.1506

https://doi.org/10.1016/j.jclepro.2018.10.075

https://doi.org/10.1016/j.foodpol.2018.08.007

The research methodology  seems underdeveloped. Methods should be described in detail. Indeed, I think the research procedure could be much more clearly described by means of a diagram also highlighting its potential and limit. 

Results must be linked to the methodology. Please define the relationship and relate your finding with the relevant literature

Conclusions are extremely succinct. I suggest to authors to propose policy implications.

Author Response

(The authors gave the same response as above.)

Round 2

Reviewer 1 Report

The second version of the paper takes into account the most relevant comments and observations. In particular, the part on methodology and the reflections about actor networks are much clearer and complete in this version.

Reviewer 2 Report

Dear authors,

you have done an excellent work. I think the paper is much stronger and deserves to be published if these few remaining issue are addressed. 

I would suggest the author to remove reference 42 since it is a working paper. 

Line 508 the authors stated "...such as nutrition education and health promotion" I would add a line where it is stated that health depends on people's diet (D'alisa et. al, 2017) https://journals.uair.arizona.edu/index.php/JPE/article/view/20782